# Combination of Everolimus and Bortezomib Inhibits the Growth and Metastasis of Bone and Soft Tissue Sarcomas via JNK/p38/ERK MAPK and AKT Pathways

**DOI:** 10.3390/cancers15092468

**Published:** 2023-04-26

**Authors:** Koichi Nakamura, Kunihiro Asanuma, Takayuki Okamoto, Takahiro Iino, Tomohito Hagi, Tomoki Nakamura, Akihiro Sudo

**Affiliations:** 1Department of Orthopaedic Surgery, Mie University Graduate School of Medicine, Tsu 514-0001, Japan; 2Department of Pharmacology, Faculty of Medicine, Shimane University, Izumo 693-8501, Japan

**Keywords:** everolimus, bortezomib, MAPK, sarcoma, metastasis, fibrosarcoma, osteosarcoma, combination, apoptosis, AKT

## Abstract

**Simple Summary:**

High-grade bone and soft tissue sarcomas are known for high recurrence and low survival rates. New drug development is costly and time-consuming. Instead, new combinations of existing drugs may offer novel anticancer strategies. We previously demonstrated that the combination of the mammalian target of rapamycin inhibitor everolimus and the proteasome inhibitor bortezomib is effective against osteosarcoma cells in a drug screening test. Building on previous research, we showed that the combination of everolimus and bortezomib exerted synergistic antiproliferative and apoptotic effects against fibrosarcoma and osteosarcoma, suppressing pulmonary metastases in osteosarcoma-bearing mice. Furthermore, we identified that everolimus and bortezomib inhibited tumor growth via the JNK/p38/ERK MAPK and AKT pathways. This combination treatment may be effective against bone and soft tissue sarcoma suppressing tumor growth and metastasis.

**Abstract:**

The combination of the mammalian target of rapamycin and proteasome inhibitors is a new treatment strategy for various tumors. Herein, we investigated the synergistic effect of everolimus and bortezomib on tumor growth and metastasis in bone and soft tissue sarcomas. The antitumor effects of everolimus and bortezomib were assessed in a human fibrosarcoma (FS) cell line (HT1080) and mouse osteosarcoma (OS) cell line (LM8) by MTS assays and Western blotting. The effects of everolimus and bortezomib on HT1080 and LM8 tumor growth in xenograft mouse models were evaluated using tumor volume and the number of metastatic nodes of the resected lungs. Immunohistochemistry was used to evaluate cleaved PARP expression. The combination therapy decreased FS and OS cell proliferation compared with either drug alone. This combination induced more intense p-p38, p-JNK, and p-ERK and activated apoptosis signals, such as caspase-3, compared with single-agent treatment. The combination treatment reduced p-AKT and MYC expression, decreased FS and OS tumor volumes, and suppressed lung metastases of OS. The combination therapy inhibited tumor growth in FS and OS and metastatic progression of OS via the JNK/p38/ERK MAPK and AKT pathways. These results could aid in the development of new therapeutic strategies for sarcomas.

## 1. Introduction

Bone and soft tissue sarcomas are rare and heterogeneous, comprising more than 50 histological subtypes [1]. These sarcomas account for approximately 1% of all malignancies [2]. Osteosarcoma (OS) is a malignant primary bone tumor that occurs mainly in adolescents and young adults [3]. Since the 1970s, systemic chemotherapy and surgical techniques have improved drastically, and the 5-year survival rate of patients with non-metastatic OS occurring in the extremities has reached 60–70% [4]. However, the prognosis is unfavorable in patients with tumors located in the axial skeleton and those with metastasis at onset (cure rate of approximately 30%) [4]. Most patients with high-grade OS present micrometastases in the lungs at diagnosis, and lung metastasis is a major cause of death [5]. Therefore, the suppression of lung metastasis is crucial for treating patients with OS.

Adult fibrosarcoma (FS) is a typical sarcoma of the soft tissues of the extremities, trunk, head, and neck. More than 80% of adult FS tumors are high-grade tumors. One in four low-grade adult FS tumors progresses to a high-grade tumor during local recurrence [6]. The 5-year survival rate in patients with high-grade FS is less than 55% [6]. Although conventional systemic chemotherapy is incorporated into the standard management of advanced sarcoma, it is often discontinued because of its considerable toxicity, poor tolerability, and limited efficacy [7].

Molecularly targeted therapy has attracted attention for the treatment of various sarcomas. Although pazopanib, trabectedin, eribulin, and immune checkpoint inhibitors have been used in sarcoma treatment, they have not been sufficiently effective to replace conventional chemotherapy [8,9,10]. Therefore, novel treatment approaches are urgently required. However, the investigation of new treatments has been particularly challenging, owing to the rarity and heterogeneity of bone and soft tissue sarcomas [10]. Compared with novel drug development, combination therapy with existing drugs is a more efficient approach to drug discovery [11]. We previously demonstrated that a combination of the mammalian target of rapamycin (mTOR) inhibitor everolimus and the proteasome inhibitor bortezomib is effective against OS cells in a drug screening test [12]. This combination showed a synergistic inhibitory effect on human OS cell lines (143B) and OS-bearing mice via apoptosis signaling pathways. The mTOR inhibitor everolimus and its analogs are used to treat various solid tumors, such as esophageal squamous cell carcinoma, lung cancer, renal cell carcinoma, and prostate cancer [13,14,15,16]. Bortezomib was the first proteasome inhibitor approved by the United States Food and Drug Administration to treat multiple myeloma and mantle cell lymphoma [17,18]. Previous studies have reported that bortezomib exerts antitumor effects in various malignancies [19,20,21,22]. However, bortezomib alone showed limited efficacy in soft tissue sarcoma in clinical trials; thus, it is recommended for use in combination with other agents [23]. The combination of mTOR inhibitors and proteasome inhibitors showed antitumor effects in myeloma, malignant peripheral nerve sheath tumors, and hepatocellular carcinoma in in vitro and in vivo studies [24,25,26].

To the best of our knowledge, there are no clinical data on the combination treatment of everolimus and bortezomib in bone and soft tissue sarcomas. Therefore, in this study, we aimed to clarify the synergistic inhibitory effects of everolimus and bortezomib on tumor growth and pulmonary metastasis of bone and soft tissue sarcomas. We investigated the synergistic effect of everolimus and bortezomib on the growth inhibition of FS and OS cells in vitro and in vivo, as well as the inhibition of lung metastases in OS using a spontaneous pulmonary metastatic mouse model. We further investigated the mechanism by which the combination of everolimus and bortezomib exerted antitumor effects on bone and soft tissue sarcomas.

## 2. Materials and Methods

### 2.1. Drugs

Everolimus and bortezomib, purchased from Wako Pure Chemical Industries Ltd. (Osaka, Japan), were used for the investigation.

### 2.2. Cells and Animals

The human FS (HT1080) and mouse OS (LM8) cell lines were used for in vitro and in vivo experiments. HT1080 cells were obtained from the Riken Cell Bank (Ibaraki, Japan). LM8 cells were kindly provided by Osaka University (Osaka, Japan; December 2014). HT1080 cells were maintained in modified Eagle medium (Gibco, Carlsbad, CA, USA) containing 10% fetal bovine serum (FBS; Invitrogen, Tokyo, Japan), whereas LM8 cells were maintained in Dulbecco’s modified Eagle medium (Gibco BRL, Grand Island, NY, USA) with 10% FBS. Cells were cultured at 37 °C in an incubator with 5% CO_2_. Sixteen 5-week-old BALB/c nu/nu mice (Charles River Laboratories Japan, Kanagawa, Japan) and sixteen 5-week-old C3H/He female mice (Charles River Laboratories Japan) were maintained in humidity- and temperature-controlled laminar flow rooms.

### 2.3. Viability Assay

HT1080 cells were seeded at a density of 1.0 × 10^4^ per well in 96-well plates and treated with 0, 5, 10, and 20 nM bortezomib and 0, 5, 10, 20, and 30 μM everolimus. LM8 cells were seeded at a density of 3.0 × 10^4^ per well in 96-well plates and treated with 0, 2.5, and 5 nM bortezomib and 0, 5, 10, 20, and 30 μM everolimus. The total amount of medium used was 100 µL/well. Each concentration was evaluated in at least six wells. After incubation with the test compounds for 12, 24, and 48 h, cell proliferation was measured using a CellTiter 96 AQ_ueous_ non-radioactive cell proliferation assay (Promega, Mannheim, Germany). The microplates were incubated for another 1.5 h at 37 ℃, and absorbance at 492 nm was measured using a microprocessor-controlled microplate reader (Infinite F200 PRO, Tecan Group Ltd., Zurich, Switzerland).

### 2.4. Calculation of Combination Index

A combination index (CI) assay was performed to evaluate whether the combination of everolimus and bortezomib enhanced the antitumor effect in the HT1080 and LM8 cell lines using CompuSyn software from ComboSyn Inc. (Paramus, NJ, USA) [27]. CI was calculated by median effect methods. Synergy was defined as CI < 1.0, antagonism as CI > 1.0, and additive effect at CI values not significantly different from 1.0.

### 2.5. Western Blot Analysis

After treatment with everolimus (5–20 μM) and bortezomib (5 nM) for 6, 12, or 24 h, HT1080 and LM8 cells were lysed with radioimmunoprecipitation buffer (Millipore-Upstate, Temecula, CA, USA) supplemented with a protease inhibitor cocktail and 0.5 mM phenylmethylsulfonyl fluoride (Roche, Mannheim, Germany). The lysates were centrifuged (12,000× *g*, 30 min, 4 °C), mixed with an SDS sample buffer (0.5 M Tris-HCl, pH 6.8, 10% SDS, 30% glycerol, 9.3% dithiothreitol, and 0.00012% bromophenol blue) at a ratio of 1:5, boiled for 5 min, and stored at −80 °C until further use. The samples were separated using 10% or 15% sodium dodecyl sulfate–polyacrylamide gel electrophoresis and transferred onto polyvinylidene difluoride membranes (Millipore Corporation, Bedford, MA, USA). After blocking the membrane with Tris-buffered saline with Tween 20 (20 mM Tris-HCl, pH 7.6, and 1% Tween-20) containing 5% nonfat dried milk for 1 h at 24 ℃, Western blot analysis was performed. The following antibodies were obtained from Cell Signaling Technology (Beverly, MA, USA) and used at the indicated dilutions: rabbit antibody against cleaved caspase-3 (Asp175; #9661; 1:1000), mouse monoclonal antibody against caspase-8 (1C12; #9746; 1:1000), rabbit monoclonal antibody against cleaved caspase-9 (Asp315; D8I9E; #20750; 1:1000), rabbit monoclonal antibody against cleaved PARP (Asp214; D64E10) XP (#5625; 1:1000), rabbit monoclonal antibody against cleaved PARP (Asp214; mouse-specific; #9544; 1:1000), rabbit monoclonal antibody against phospho-p44/42 MAPK (extracellular signal-regulated kinase 1/2) (ERK1/2) (Thr202/Tyr204; D13.14.4E) XP (#4370; 1:1000), rabbit polyclonal antibody against ERK1/2 (#9102; 1:1000), rabbit antibody against phospho-stress-activated protein kinases (SAPK)/phospho-JNK (Thr183/Tyr185; #9251; 1:1000), rabbit polyclonal antibody against JNK (#9252; 1:1000), rabbit monoclonal antibody against phospho-p38 MAPK (Thr180/Tyr182; D3F9; #4511; 1:1000), rabbit monoclonal antibody against p38 MAPK (#9212; 1:1000), rabbit monoclonal antibody against c-MYC proto-oncogene basic helix–loop–helix transcription factor (MYC; D84C12; #5605; 1:1000), rabbit monoclonal antibody against survivin (#2808; 1:1000), rabbit monoclonal antibody against MCL-1 apoptosis regulator, BCL2 family member (MCL-1; #5453; 1:1000), rabbit antibody against phospho-AKT serine/threonine kinase 1 (AKT1; (#4060; 1:1000), and rabbit antibody against AKT (#9272; 1:1000). Rabbit polyclonal antibody against BCL2-interacting killer (BIK) was obtained from Abcam (ab52182; 1:500; Abcam, Oxford, UK). β-Actin (ab8229; 1:500) protein was used as a loading control and was purchased from Abcam. After hybridization with a horseradish peroxidase-conjugated secondary antibody, the bands were visualized using the ECL Prime Western blotting system (Cytiva, Little Chalfont, UK).

### 2.6. In Vivo Tumorigenesis Assay

For xenografting, 2 × 10^6^ each of HT1080 and LM8 cells in 0.2 mL of phosphate-buffered saline (PBS) was subcutaneously injected into the backs of the BALB/c nu/nu and C3H/He mice, respectively, using a 26-gauge needle. In all mice, enlargement of the tumors was observed within one week after inoculation. The tumor size was measured with calipers twice a week, and the tumor volume (cm^3^) was calculated using the ellipsoid formula:*V* = *l* × *w*^2^ × 0.52(1)
where *l* is the length, *w* is the width, and *V* is the volume, as described previously [11]. The body weights of the mice were measured twice per week. Sixteen mice of each type—BALB/c nu/nu and C3H/He—were randomly assigned to four groups (four mice per group): vehicle alone (as a control), everolimus alone, bortezomib alone, and a combination of everolimus and bortezomib. Everolimus was dissolved in dimethyl sulfoxide (DMSO) and diluted in saline to a final concentration of DMSO of 0.1% or less, and bortezomib was dissolved in saline solution. Seven days after cell implantation, the mice were treated with an intraperitoneal injection of bortezomib at a dose of 0.5 mg/kg in 200 μL of 0.9% saline solution once a week (on day 1), and everolimus was administered orally at a dose of 1 mg/kg three times a week (on days 1, 4, and 6). This schedule was considered to be one cycle (Figure 1). The dosages of everolimus and bortezomib were based on previous studies [12]. The mice treated with both everolimus and bortezomib were kept on the same schedule as that described for the single-drug treatment. Control mice received 200 μL of saline orally (three times a week) and 200 μL of saline by intraperitoneal injection (once a week), including the same concentration of DMSO as the treatment groups. The treatment was stopped after five cycles. BALB/c nu/nu and C3H/He mice were euthanized by an overdose of pentobarbital on day 36.

### 2.7. Histopathological Analyses

Mice were euthanized for histopathological analysis. Tumor tissue sections of HT1080 were subjected to immunohistochemical analyses. Tumor blocks were formalin-fixed, paraffin-embedded, and sliced into 4 μm-thick sections. These sections were then deparaffinized in xylene and rehydrated using decreasing ethanol concentrations (100%, 95%, 85%, and 75%), followed by incubation in 3% H_2_O_2_ for 30 min in the dark at 24 °C to eliminate endogenous peroxidase activity. Antigen retrieval was performed by heating the sections for 10 min in citrate buffer (pH 6.0) using an autoclave sterilizer. The sections were allowed to cool to 24 °C for 60 min, then rinsed three times for 5 min with fresh PBS. Thereafter, the slides were preincubated with healthy bovine serum albumin diluted in PBS (pH 7.4) for 15 min at 37 °C, then incubated overnight at 24 °C with primary antibodies specific to cleaved PARP (#5625, 1:50; Cell Signaling Technology). After three rinses in fresh PBS, the slides were incubated with a horseradish-peroxidase-coupled secondary antibody (#414321, Nichirei Biosciences Inc., Tokyo, Japan) for 40 min at 24 °C. Following three additional washes, all specimens were stained with a 3,3’-diaminobenzidine (DAB) substrate. Finally, the sections were rinsed in distilled water and counterstained with Mayer’s hematoxylin according to the manufacturer’s instructions. Cytoplasmic staining for cleaved PARP was defined as positive, and the percentage of positively stained cells among the total number of malignant cells was scored [28]. We randomly counted eight fields to score cleaved PARP using Olympus cellSens ver.1.18 software and a BX50 microscope equipped with a DP74 camera (Olympus, Tokyo, Japan). 

### 2.8. Antimetastatic Effect of Bortezomib and Everolimus In Vivo

The antimetastatic potential of everolimus and bortezomib against LM8 cells was evaluated in vivo. C3H/He mice were euthanized 36 days after the initial treatment. The lungs were carefully excised, fixed with formalin, embedded in paraffin, sectioned, and stained with hematoxylin and eosin for histological observation. Lung metastases were determined by counting the metastatic nodes at the maximum plane of the lungs.

### 2.9. Statistical Analyses

All statistical analyses were performed using the EZR graphical user interface (Saitama Medical Center, Jichi Medical University, Saitama, Japan) for R (The R Foundation for Statistical Computing, Vienna, Austria), which is a modified version of the R Commander designed to add statistical functions frequently used in biostatistics [29].

Data are expressed as the mean ± standard error of the mean. One-way analysis of variance followed by Tukey’s test was used for multiple comparisons. The immunohistostaining rates among the four groups were compared using the Kruskal–Wallis chi-squared test. Differences were considered statistically significant at *p* < 0.05.

## 3. Results

### 3.1. Evaluation of the Synergistic Effect of Everolimus and Bortezomib

The antiproliferative effects of the mTOR inhibitor everolimus and proteasome inhibitor bortezomib were investigated in HT1080 and LM8 cells. HT1080 and LM8 cell growth was markedly inhibited by each drug alone in a dose- and time-dependent manner (Figure 2A,B). The CI values for the combination of everolimus and bortezomib were less than 1.0, demonstrating synergy between everolimus and bortezomib in HT1080 and LM8 cells. Synergistic antiproliferative effects were observed in HT1080 cells treated with 5 nM bortezomib and 5 μM everolimus (Figure 2C) and, to a greater extent, in LM8 cells treated with 2.5 nM bortezomib and 10 μM everolimus (*p* < 0.01; Figure 2D). The inhibition rates of HT1080 and LM8 cells in the combination treatment group were 75.1% and 81.1%, respectively, compared with the control group at the 48 h time point.

### 3.2. Cell Signaling Mechanism

HT1080 cells were treated with or without 5, 10, or 20 μM everolimus and 5 nM bortezomib for 6, 12, and 24 h (Figure 3A). Combination therapy with everolimus and bortezomib enhanced the levels of cleaved caspase-3 and cleaved PARP at 6, 12, and 24 h (Figure 3A top), indicating that the combination treatment triggered strong apoptotic signals. The enhanced activation of caspase-9 indicates the involvement of intrinsic apoptotic pathways. The expression of caspase-8 was not increased in the combination treatment group compared with that in the single-treatment or control groups. This indicated that the combination treatment did not affect the extrinsic apoptotic pathways. Enhanced levels of p-JNK and p-p38 are associated with apoptosis induction (Figure 3A middle). However, the level of MCL-1, a BH3 antiapoptotic protein, was enhanced by combination treatment (Figure 3A, bottom). The level of BIK, a proapoptotic protein, was unaffected by combination treatment (Figure 3A top). These results indicate that MCL-1 and BIK do not play a key role in promoting the apoptosis cascade in HT1080 cells.

The expression of MYC, an important protein for cell proliferation, was reduced after combination treatment for 24 h (Figure 3A bottom). The expression of survivin, an inhibitor of apoptosis proteins, was dose-dependently reduced by everolimus and increased following bortezomib treatment (Figure 3A bottom). At 6 h, the level of p-AKT was reduced with combination treatment (Figure 3A bottom). A dose-dependent enhancement of p-ERK activation, acting as a proapoptotic protein, was observed with combination treatment at 6 h and 12 h (Figure 3A middle). After 24 h, the levels of p-ERK were dose-dependently reduced by everolimus treatment. In contrast, treatment with bortezomib enhanced p-ERK levels at 6, 12, and 24 h (Figure 3A middle). These results indicate that the combination treatment triggered apoptosis via the p-ERK pathway.

Combination therapy enhanced the levels of p-JNK and p-p38 in this study (Figure 3A,B, middle), indicating that the combination treatment enhanced apoptosis via the JNK/p38 MAPK pathway. The expression of MYC was reduced after combination treatment for 24 h (Figure 3A,B bottom). Survivin expression was unaffected after combination treatment (Figure 3A,B bottom). The expression of p-AKT was reduced with combination treatment in a dose-dependent manner (Figure 3A,B bottom), indicating that the combination treatment interfered with cell proliferation via the AKT and MYC pathways.

### 3.3. In Vivo Effects of Everolimus and Bortezomib

To analyze the synergistic effects of everolimus and bortezomib in vivo, we examined time-dependent changes in tumor volumes in HT1080 and LM8 xenografts (Figure 4A,B). In the absence of everolimus and bortezomib, the growth of HT1080 and LM8 tumors was aggressive. Single treatments with everolimus or bortezomib slightly reduced tumor growth (but no significant difference was observed between the single treatment and the control groups), whereas combination treatment with everolimus and bortezomib significantly suppressed HT1080 and LM8 tumor growth relative to the control group on day 36 (*p* = 0.018 and *p* = 0.004, respectively). No weight loss of over 20% compared with before treatment was observed in any group during the treatment (Figure 4C,D).

The expression of cleaved PARP, the fundamental hallmark of apoptosis, was observed in tumor tissues in the single and combination treatment groups (Figure 5), suggesting that the inhibition of tumor growth was the result of apoptosis.

To evaluate the antilung-metastatic effect of everolimus and bortezomib in vivo, we used a spontaneous pulmonary metastasis mouse model and evaluated the number of lung nodules in LM8 xenografts. The number of pulmonary nodules was high in the untreated groups (Figure 6A). A single treatment with everolimus or bortezomib slightly reduced the number of pulmonary nodules, whereas combination treatment significantly reduced the number of pulmonary nodules (*p* = 0.0351; Figure 6B).

## 4. Discussion

### 4.1. The JNK/p38/ERK MAP Kinase Pathway

MAPKs belong to a large family of serine/threonine kinases, which are major components of signaling pathways involved in cell proliferation, differentiation, and death. Currently, there are three known MAPKs: ERK1/2, SAPK/JNK, and p38. JNKs, p38, and ERK form the last tier of the three-tier kinase module consisting of MAP3K, MAP2K, and MAPK [30] (Figure 7). Apoptosis signal-regulating kinase 1 (ASK1) is an MAP3K that activates the penultimate MAP2K. MAP2K activates JNKs and p38 kinases via phosphorylation [31]. ASK1 is activated by various stresses and stimuli, such as tumor necrosis factor-α, reactive oxygen species (ROS), endoplasmic reticulum stress, and lipopolysaccharide. JNK and p38 are strongly activated by ROS, leading to caspase cleavage and apoptosis in HeLa cells [32]. JNK signaling plays a critical role in apoptosis through death receptor-initiated extrinsic and intrinsic mitochondrial pathways [33].

The proteasome regulates cellular homeostasis via the degradation of dysfunctional intracellular proteins and the rapid turnover of regulatory proteins via the ubiquitin–proteasome pathway [34,35]. Bortezomib causes protein overload, endoplasmic reticulum stress, and ROS accumulation, leading to caspase cleavage and induction of apoptosis via the intrinsic mitochondrial pathway [21,34]. Bortezomib induces apoptosis via the JNK/p38 MAPK pathway in esophageal squamous cell carcinoma, multiple myeloma cells, and glioblastoma cells [22,36,37]. The RAF/MEK/ERK signaling pathway regulates diverse cellular processes such as proliferation, differentiation, motility, and survival [38]. Depending on the cell type and the nature of the stimuli, ERK activation is associated with the intrinsic apoptotic pathway, characterized by the release of cytochrome C from mitochondria and initiator caspase-9 activation, or with the extrinsic apoptotic pathway [39,40]. ERK acts upstream of caspase-3 in cisplatin-induced cell death [39]. The mechanism underlying ERK1/2-mediated cell death is not fully elucidated. Recent studies reported that the expression of dual-specificity phosphatases (DUSPs) determines the pro- versus antiapoptotic function of ERK in cancer [38,41].

In the present study, combination treatment synergistically induced JNK/p38/ERK activation in HT1080 and LM8 cells. This combination therapy may elicit intrinsic mitochondrial apoptosis of FS and OS cells via the JNK/p38/ERK MAPK pathway.

### 4.2. The AKT Pathway

Ras/Raf/MAPK and PI3K act downstream of the receptor tyrosine kinase pathway (Figure 7). Aberrant regulation of the PI3K/AKT signaling pathway plays a pivotal role in tumorigenesis, metastasis, and resistance to standard chemotherapy [42]. Thus, treatment strategies targeting PI3K/AKT/mTOR signaling have received attention [43]. PIP3 induces the activation of phosphoinositide-dependent kinase-1 (PDK1) and downstream targets of AKT [44]. Phosphatase and tensin homolog (PTEN) is a lipid phosphatase that antagonizes the action of PI3K by dephosphorylating PIP3 to generate PIP2. PTEN functions as a tumor suppressor protein that suppresses AKT [45], ultimately affecting broad downstream targets that govern endothelial cell proliferation, invasion, apoptosis, and angiogenesis [17,46]. Angiogenesis, a common hallmark of cancer, is crucial for metastasis [43]. Membrane-bound AKT is fully activated by PDK1 and mTOR phosphorylation; p-AKT may phosphorylate a range of substrates, resulting in cellular growth, survival, and proliferation through various mechanisms [47,48]. In addition, ERK activation can promote cell death by suppressing the AKT signaling pathway in renal cells [47]. Cross talk between ERK and AKT signaling pathways is reported in lung cancer [49,50,51] (Figure 7).

Everolimus or its analog decreases PI3K, AKT, and mTOR expression levels in esophageal squamous cell carcinoma and breast cancer cells [14,25,52]. The mTOR inhibitor everolimus shows an antitumor effect in malignant peripheral nerve sheath tumor-bearing mouse models by promoting a cytostatic effect [53]. 

In this study, the combination of everolimus and bortezomib synergistically reduced p-AKT expression, which might contribute to the suppression of lung metastasis in OS-bearing mice via the mTORC1-MYC pathway, in addition to synergistic antiproliferative effects on FS- and OS-bearing mice.

### 4.3. MYC

mTOR complex 1 (mTORC1) is a central sensor of amino acid signaling. mTORC1 signaling regulates a variety of metabolic pathways by activating transcription factors, including the oncogenic transcription factor MYC. mTORC1 activates MYC to promote hepatocellular carcinoma tumorigenesis by modulating methionine metabolism [54]. MYC can provide an advantage to cancer cells by promoting proliferation and angiogenesis, helping to evade the immune response [55]. Inhibition of MYC leads to cell cycle arrest and apoptosis in several types of tumor cells [56,57,58]. In bladder cancer, the expression of MYC is activated by AKT-mTOR signaling [59]. In Burkitt’s lymphoma, bortezomib downregulates MYC expression [21]. In this study, bortezomib downregulated MYC expression in HT1080 and LM8 cells. Combination treatment reduced MYC expression in HT1080 and LM8 cells, leading to the inhibition of cell proliferation and lung metastasis.

### 4.4. Synergic Mechanism

We successfully demonstrated synergic effects of the combination of everolimus and bortezomib. In this study, we concluded that the main factors were p38, JNK, AKT, and MYC. The combination therapy strongly activated p38 and JNK, inducing apoptosis. This combination therapy also inhibited AKT and MYC, inhibiting cell proliferation (Figure 7). These effects contributed to the suppression of pulmonary metastasis in OS-bearing mice. Additionally, p-ERK reduces p-AKT expression through cross talk between p-AKT and p-ERK [47,50]. In this study, the combination therapy upregulated p-ERK expression, which may be related to p-AKT expression. ERK, known as a double-edged sword for cell proliferation, may be one of the key factors for anticancer therapy (Figure 7) [38].

## 5. Conclusions

Many researchers have determined various molecular targets for cancer treatment, and new drugs for these targets are currently being developed. However, new drug development is costly and time-consuming. Instead, new combinations of existing drugs may be novel strategies for cancer treatment. In this study, we showed that the combination of everolimus and bortezomib exerted synergistic antiproliferative and apoptotic effects against FS and OS, suppressing pulmonary metastases in OS-bearing mice via the JNK/p38/ERK MAPK and AKT pathways. In addition, the occurrence of metastases was reduced in a spontaneous pulmonary metastasis model. This model closely mimics the metastasis of patients with sarcoma in clinical settings rather than an experimental metastasis model. We suggest that this combination treatment may be effective against bone and soft tissue sarcoma by suppressing tumor growth and metastasis.

## Figures and Tables

**Figure 1 cancers-15-02468-f001:**
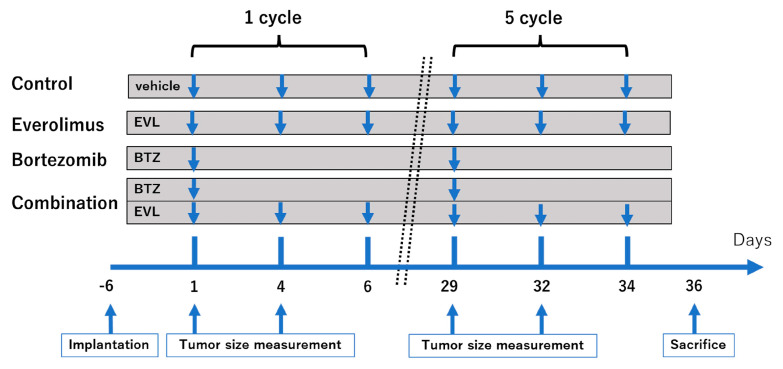
Treatment schedule. Tumor-bearing mice (n = 4 per group) were orally administered vehicle (three times a week), everolimus (1.0 mg/kg, three times a week) or intraperitoneally injected with vehicle (once a week), bortezomib (0.5 mg/kg, once a week).

**Figure 2 cancers-15-02468-f002:**
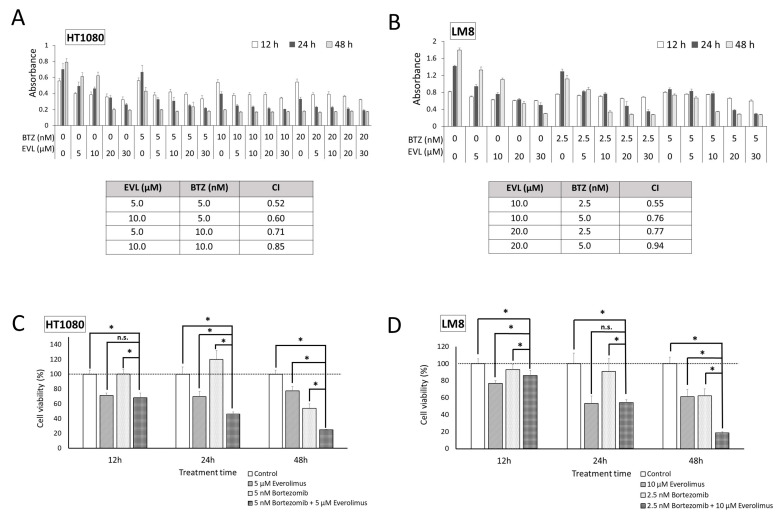
Dose–response data of HT1080 and LM8 cells to everolimus and bortezomib to assess cell viability. Everolimus was used at a concentration of 5–30 µM, and bortezomib was used at a concentration of 2.5–20 nM. The effects of combination treatment with everolimus and bortezomib were analyzed in HT1080 (**A**) and LM8 (**B**) cells at 12, 24, and 48 h. The combination index (CI) values for everolimus (EVL) + bortezomib (BTZ) were < 1, indicating a synergistic effect between EVL and BTZ in treating HT1080 and LM8 cells. (**C**) Remarkable synergistic antiproliferative effects were observed in the presence of bortezomib (5 nM) and everolimus (5 μM) in HT1080 cells. (**D**) Remarkable synergistic antiproliferative effects were observed in the presence of bortezomib (2.5 nM) and everolimus (10 μM) in LM8 cells. BTZ: bortezomib; EVL: everolimus; n.s.: not significant; * *p* < 0.01.

**Figure 3 cancers-15-02468-f003:**
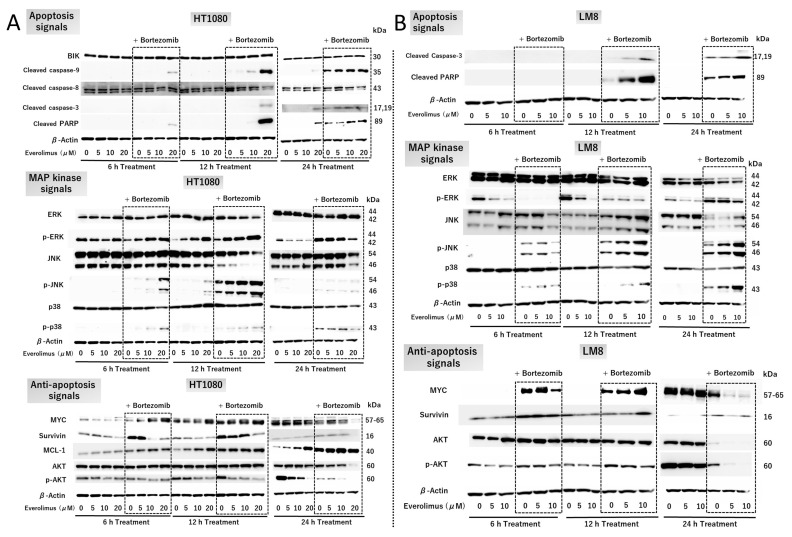
Cell signaling with everolimus and bortezomib treatment. HT1080 and LM8 cells were used for Western blot analysis. (**A**) HT1080 cells were treated with 0–20 µM everolimus with and without 5 nM bortezomib for 6, 12, and 24 h. (**B**) LM8 cells were treated with 0–10 µM everolimus with and without 2.5 nM bortezomib for 6, 12, and 24 h. BIK: BCL2-interacting killer; ERK: extracellular signal-regulated kinase; p-JNK: phospho-c-Jun N-terminal kinase; PARP: poly (ADP-ribose) polymerase. The original Western blot figures can be found in Appendix A.

**Figure 4 cancers-15-02468-f004:**
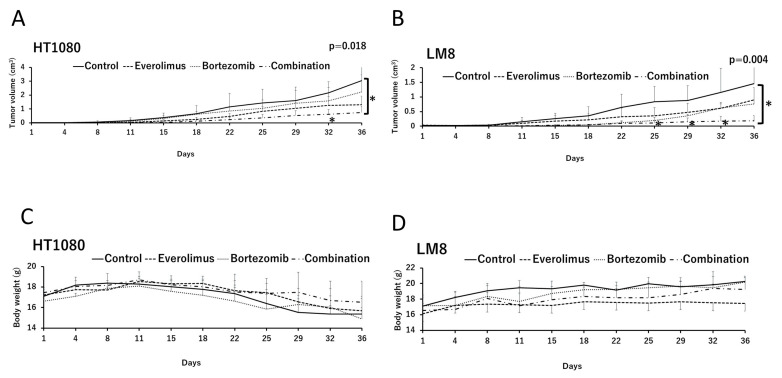
In vivo antitumor effects of everolimus and bortezomib in athymic mice with HT1080 and LM8 xenografts treated with the vehicle (control), single therapy with bortezomib or everolimus, or combination therapy of everolimus and bortezomib (combination). The effects of everolimus and bortezomib on tumor growth were assayed in (**A**) HT1080 and (**B**) LM8 cells. Data are represented as the mean tumor volumes ± standard error; *n* = 4 mice per group. * *p* < 0.05; combination vs. control on days 32 and 36 in HT1080 xenografted mice and on days 25, 29, 32, and 36 in LM8 xenografted mice using one-way analysis of variance. Body weight of mice with (**C**) HT1080 and (**D**) LM8 xenografts in the different treatment groups. The body weights of the mice were measured twice a week.

**Figure 5 cancers-15-02468-f005:**
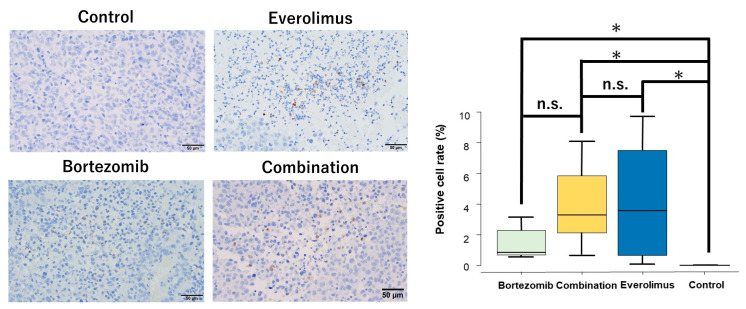
Representative tissue sections of immunohistochemical analysis of cleaved PARP in HT1080 xenografts. Original magnification: ×400. The mean positive cell rates of cleaved PARP in the randomized field of view were 3.3% (combination), 3.6% (everolimus), 0.85% (bortezomib), and 0% (control). The positive rates of the combination, everolimus, and bortezomib groups were significantly higher than those of the control group (*p* = 0.0041, *p* = 0.0059, and *p* = 0.0041, respectively). * *p* < 0.01.

**Figure 6 cancers-15-02468-f006:**
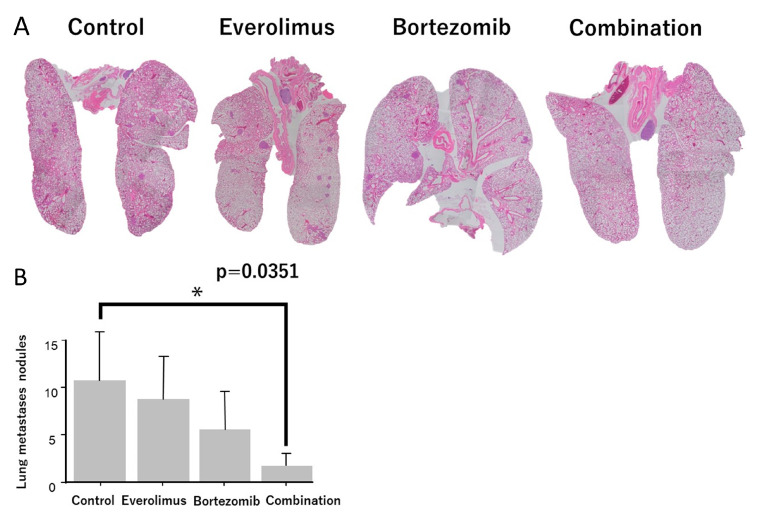
Evaluation of lung metastases between control and treatment groups. (**A**) Hematoxylin and eosin staining was performed, and the metastatic foci were counted. (**B**) The number of metastatic foci in the combination treatment group was significantly lower than that in the control group. * *p* < 0.05.

**Figure 7 cancers-15-02468-f007:**
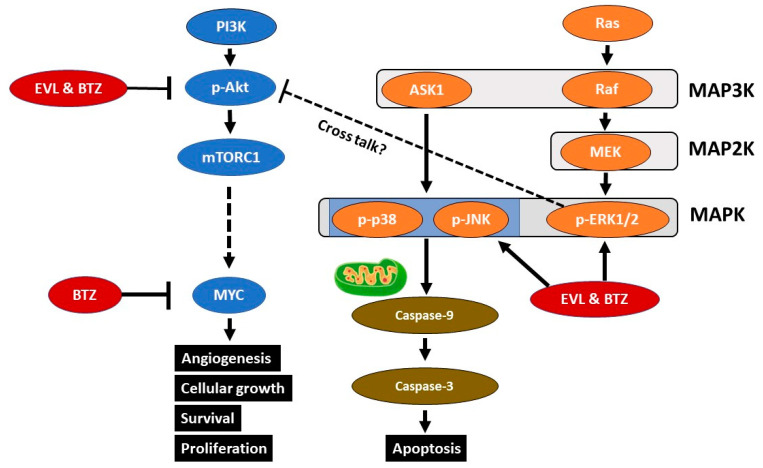
Signaling cascades downstream of the tyrosine kinase pathway. The combination of everolimus and bortezomib (EVL and BTZ) results in activation of p-ERK1/2, p-p38, and p-JNK, triggering intrinsic mitochondrial apoptosis. The combination therapy downregulated p-AKT expression, contributing to a reduction in MYC expression via mTOR complex 1 (mTORC1).

## Data Availability

The data generated in this study are available upon request from the corresponding author.

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
