# Peer review of "Combination of Everolimus and Bortezomib Inhibits the Growth and Metastasis of Bone and Soft Tissue Sarcomas via JNK/p38/ERK MAPK and AKT Pathways"

_cancers, 2023, doi:10.3390/cancers15092468_

Round 1

Reviewer 1 Report

In the manuscript from Koichi Nakamura et. al. are authors describing new options for combination treatment of Bone and Soft Tissue Sarcomas. The authors describing the combination of Bortezomib and Everolimus. The findings are not very surprising as this has been previously shown on osteosarcoma cells. 

 In regards of the data I have several main issues which has to be corrected by the authors.

In regards to:

Figure 2

1.    The viability data have to be normalized to the untreated control to be able to better judge the claim of the synergism, which is further connected to the wording “remarkable” in the description of the figure.

2.    If claiming synergy beween drugs, proper statistics has to be use, like calculating the combinatorial index e.g.

 Figure 3

1.    All blots showing phorphorylation or cleavage has to be complemented with the non-phospho or non-cleaved form of the particular protein

The discussion is only very short review of the pathways involved, while by no means setting the results in to the prospective to current knowledge or results. I am also missing the explanation of the mechanism of action, why it works.

I would propose to authors to shorter the manuscript and try to submit as a short report, or letter rather than full manuscript.

Author Response

Prof.  Samuel C. Mok

Editor-in-Chief

Cancers

Resubmission of Manuscript NO. 2249367

Dear Dr. Mok,

Thank you for your email dated 11 March 2023 enclosing the reviewers’ comments. We

have carefully reviewed the comments and have revised the manuscript accordingly. Our

responses are given in a point-by-point manner below. The changes incorporated in the manuscript are marked in red font.

We hope that we have addressed the reviewer’s concerns satisfactorily and that the revised version is now suitable for publication in Cancers.

Thank you for your consideration. We look forward to hearing from you in due course.

Sincerely,

Kunihiro Asanuma, M.D., Ph D.

Department of Orthopedic Surgery, Mie University Graduate School of medicine, 2-174 Edobashi, Tsu, Mie Prefecture, Japan. 514-8507

TEL: +81-59-231-5022, FAX: +81-59-231-5211

e-mail: kasanum@gmail.com

Thank you for your review of our paper. We have addressed your comments below.

  1. [Figure 2; The viability data have to be normalized to the untreated control to be able to better judge the claim of the synergism, which is further connected to the wording “remarkable” in the description of the figure.;]

Response:

We agree with you and have incorporated this suggestion. We have changed Figure 2C, 2D to explain synergism by normalizing to the untreated control data (p. #6, lines, #237).

  1. [Figure 2; If claiming synergy between drugs, proper statistics has to be use, like calculating the combinatorial index e.g.]

Response:

Thank you for your suggestion. We have supplemented the 2.4. Calculation of combination index section. (p.#3, lines #119-#124). We added new tables and explanation in Figure 2 (p. #6, lines, #229-#231, #237, #241-243).

The combination index (CI) was calculated by the CompuSyn software from ComboSyn Inc. (New Jersey, USA). Synergy was defined as CI < 1.0, antagonism as CI > 1.0, and additive effect at CI values not significantly different from 1.0. Most synergistic antiproliferative effects were observed in the presence of bortezomib (5 nM) and everolimus (5 μM) in HT1080 (CI=0.52), in the presence of bortezomib (2.5 nM) and everolimus (10 μM) in LM8 (CI=0.55).

  1. [Figure 3 All blots showing phosphorylation or cleavage has to be complemented with the non-phospho or non-cleaved form of the particular protein]

Response: As you suggested, it would be ideal to investigate at all non-phosphorylated and truncated protein expressions. However, investigating this requires the cost of purchasing antibodies and laborious experiments and we could not finish data making by the end of date of resubmission. This time, we performed WB using antibodies in previous experiments on human osteosarcoma cells (143B) [1]. There are some papers that discussed the mechanism of action based on the expression of cleaved caspase-3, cleaved-caspase-9, p-Akt, p-JNK, p-p38, and p-ERK, not investigated non-phospho or non-cleaved form of the particular protein [1], [2]. In this study, we did not examine the expression of non-phosphorylated and non-cleavage proteins.

Reference

  1. Asanuma, K.; Nakamura, T.; Nakamura, K.; Hagi, T.; Okamoto, T.; Kita, K.; Matsuyama, Y.; Yoshida, K.; Asanuma, Y.; Sudo, A. Compound library screening for synergistic drug combinations: mTOR inhibitor and proteasome inhibitor effective against osteosarcoma cells. Anticancer Res. 2022, 42, 4319-4328.
  2. Stulpinas, A.; Sereika, M.; Vitkeviciene, A.; Imbrasaite, A.; Krestnikova, N.; Kalvelyte, A.V. Crosstalk between protein kinases AKT and ERK1/2 in human lung tumor-derived cell models. Front. Oncol. 2022, 12, 104552
  3.  
  4.  [The discussion is only very short review of the pathways involved, while by no means setting the results in to the prospective to current knowledge or results. I am also missing the explanation of the mechanism of action, why it works.]

Response:

Thank you for providing these insights. We have included a new Figure 7 to further illustrate the mechanism of action (p.#12 #385-#389). Besides, we added some explanation to understand easily the action mechanism of the combination therapy (p.#12 #383, p.#12 #383, p.#12 #398-403). The 4.4. Synergic mechanism section has been added to make it easier for readers to understand the mechanism of combination therapy (p.12-13, lines #406-#416).

In this combination therapy, main factors were p38, JNK, AKT and MYC. The combination therapy strongly activated p38 and JNK, resulting in apoptosis induction. Besides, this combination therapy also inhibited AKT and MYC, leading to the inhibition of cell proliferation. These effects contributed to the suppression of pulmonary metastasis in OS-bearing mice. Additionally, p-ERK reduces the expression of p-Akt through crosstalk between p-Akt and p-ERK [3] [4]. In this study, the combination therapy upregulated p-ERK expression, which may relate to the expression of p-Akt. ERK, known as a double-edged sword for cell proliferation, may be one of the key factors for anti-cancer therapy [3].

In conclusion, the main factors were p38, JNK, AKT and MYC. The combination therapy strongly activated p38 and JNK, resulting in apoptosis induction. Besides, this combination therapy also inhibited AKT and MYC, leading to the inhibition of cell proliferation.

Reference

  1. Sinha, D.; Bannergee, S.; Schwartz, J.H.; Lieberthal, W.; Levine, J.S. Inhibition of ligand-independent ERK1/2 activity in kidney proximal tubular cells deprived of soluble survival factors up-regulates Akt and prevents apoptosis. The Journal of biological chemistry. 2004, 279, 10962–10972.
  1. Stulpinas, A.; Sereika, M.; Vitkeviciene, A.; Imbrasaite, A.; Krestnikova, N.; Kalvelyte, A.V. Crosstalk between protein kinases AKT and ERK1/2 in human lung tumor-derived cell models. Frontiers in oncology. 2022, 12, 1045521.

Reviewer 2 Report

In this article by Nakamura et al., has described the combinatorial effect of Everolimus and Bortezomib co-treatment on growth inhibition and metastasis inhibition of soft tissue sarcomas. As a mechanism, they have delineated JNK/p38/ERK and AKT pathways that are activated and inhibited, respectively upon the co-treatment of these agents. The article is well written, easy flowing with interesting result. This should be accepted in Cancers with minor improvements.

Can the author present a quantitative analysis (quantifying the cleaved PARP signal) of figure 5?

Author Response

Prof.  Samuel C. Mok

Editor-in-Chief

Cancers

Resubmission of Manuscript NO. 2249367

Dear Dr. Mok,

Thank you for your email dated 11 March 2023 enclosing the reviewers’ comments. We have carefully reviewed the comments and have revised the manuscript accordingly. Our responses are given in a point-by-point manner below. The changes incorporated in the manuscript are marked in red font.

We hope that we have addressed the reviewer’s concerns satisfactorily and that the revised version is now suitable for publication in Cancers.

Thank you for your consideration. We look forward to hearing from you in due course.

Sincerely,

Kunihiro Asanuma, M.D., Ph D.

Department of Orthopedic Surgery, Mie University Graduate School of medicine, 2-174 Edobashi, Tsu, Mie Prefecture, Japan. 514-8507

TEL: +81-59-231-5022, FAX: +81-59-231-5211

e-mail: kasanum@gmail.com

Thank you for your review of our paper. We have addressed your comments below.

[Can the author present a quantitative analysis (quantifying the cleaved PARP signal) of figure 5?]

Response: Thank you for your suggestion. We randomly counted eight fields for scoring cleaved PARP. We have reflected this comment by adding tables and explanations Figure 5 (p.#5, lines #204-#208, #221-#223, p.#9, #311-#317). The mean positive cell rates of cleaved PARP were 3.3% (Combination), 3.6% (Everolimus), 0.85% (Bortezomib), and 0% (Control). Significant differences were observed between the control group and the single or combination groups (p<0.05).

Round 2

Reviewer 1 Report

although the author did improve the manuscript, the control for the non phospho forms are still missing (I could accept the missing controls for the cleaved form of caspases). I also understand the point that the antibodies are relativelly expensive, but I think the research should be perfomed anyway by using propper controls and proper design of the experiments.

I will leave the decission on the editor.

Author Response

Reviewer #1 comments (after revision):

To whom it may concern, although the author did improve the manuscript, the control for the non phospho forms are still missing (I could accept the missing controls for the cleaved form of caspases). I also understand the point that the antibodies are relatively expensive, but I think the research should be performed anyway by using proper controls and proper design of the experiments.

I will leave the decision on the editor.

Academic editor:

Reviewer 1 point out that some controls are still missing.
The paper can't be published without these missing controls, so that my decision is Major revision before reconsideration.

Response:

We agreed that control experiments are crucial for the robustness of our results. JNK, ERK, AKT, and p38 are key factors in this study, so we performed control experiments; the corresponding results have been added to Figure 3A, 3B (p. #8, lines, #280). However, cyclin D was not mentioned in the discussion because it showed a low impact on the antitumor efficacy of everolimus and bortezomib in this study. Therefore, we removed P-Cyclin D experiments from the manuscript and omitted the Cyclin D control experiment (p. #8, lines, #280). We changed Figure 3A, 3B bottom title Anti-apoptosis signals from Anti-apoptosis and cell cycle. This does not affect the conclusions of this paper.

There was no change in the experiment and statistical data; however, there was an error in the notation, so we have corrected it (Figure 2C HT1080 treatment 12 h treatment EVL vs. EVL+BLZ: * → n.s.). (p. #6, lines, #238)

We hope the revised version is now suitable for publication and look forward to hearing from you in due course.

Sincerely,

Kunihiro Asanuma

Round 3

Reviewer 1 Report

I am fullly satisfied.

Thanks